# Pricing and Service Level Decisions under a Sharing Product and Consumers' Variety-Seeking Behavior

**Zhenfeng Liu** [1,2], **Jian Feng** [3] **and Bin Liu** [1,*]

[1]  School of Economics and Management, Shanghai Maritime University, Shanghai 201306, China;
   zfliu@shmtu.edu.cn
[2]  Economics and Management School, Wuhan University, Wuhan 430072, China
[3]  China Institute of FTZ Supply Chain, Shanghai Maritime University, Shanghai 201306, China;
   jfeng@shmtu.edu.cn
[*]  Correspondence: liubin@shmtu.edu.cn

**Abstract:** We developed a two-period duopoly model to show how consumers' variety-seeking behavior affects the pricing and service level decisions of a traditional product and a sharing product. Our analysis revealed that, without considering the consumers' variety-seeking behavior, the traditional product attracted consumers with a high level of service and high price, while the sharing product attracted consumers with a low level of service and low price. When we only considered variety-seeking behavior and did not adjust the service level, the product with the low level of service benefited from the consumers' variety-seeking behavior, while the product with the high level of service lost profits. When we considered the variety-seeking behavior and adjusted the service level as well as the price, the sharing product was attractive to variety-seeking consumers and it gained a greater competitive advantage over the traditional product. For two periods, the number of variety-seeking consumers who switched from buying traditional products to buying sharing products was greater than those who switched from buying sharing products to buying traditional products. Furthermore, we found that when the consumers' variety-seeking behavior was not obvious, the number of consumers shifting from the traditional product increased monotonically. In contrast, when the variety-seeking behavior was obvious, the number of consumers shifting from the traditional product decreased monotonically.

**Keywords:** sharing economy; variety-seeking behavior; consumer preference; pricing strategy; service level; game theory

---

## 1. Introduction

The Sharing Economy (SE) has seen phenomenal growth in recent years with the development of online technological advances and the spread of mobile communication devices [1,2]. Consumers have so far enjoyed the services offered by service sharing platforms, such as Airbnb, Uber, TaskRabbit, and DiDi. For example, people enjoy the convenience of taking an Uber and the user-friendliness of accommodations via Airbnb. Therefore, unlike traditional established industries, a SE mainly provides a personalized, memorable, convenient, user-friendly, interpersonal, unique, and innovative experience. In addition, the sharing products' experiences increase product diversity and provide consumers with more purchasing options besides those of traditional products. Consequently, the SE influences and promotes consumers' variety-seeking behavior [3]. This behavior is widely observed in existing SE literature. Tussyadiah and Pesonen (2015) [4], Guttentag et al. (2018) [5], and Lin et al. (2019) [6] have studied the switching phenomenon in which consumers choose Airbnb instead of traditional hotels. This behavior, moreover, induces competition between traditional products and

sharing products. For instance, in recent studies, the cannibalization by Airbnb of the hotel sector's market and profits has been discussed. The Hotel Association of New York City estimated that in the 12 months ending in August 2015, Airbnb caused a direct loss of $451 million for New York City hotels. Zervas et al. (2017) found that in areas where Airbnb is most popular, the revenue of the most vulnerable hotels has decreased by approximately 8–10% over the five years leading up to 2017 [7]. Farronato and Fradkin (2018) found that Airbnb reduced variable hotel profits from accommodations by up to 3.7% in top U.S. cities [8].

However, in the face of the loss of profits due to consumers' variety-seeking behavior, we focus on how this behavior affects consumers' purchasing decisions and how variety-seeking customers can be managed both in the sharing market and in the traditional market. We analyze the impact of consumers' variety-seeking behavior on firms' decisions. Firms may adopt some decisions, such as price discrimination [9], price pre-commitment [10], and adjusting quality level [11], to manage variety-seeking customers. We posed the following research question: How will consumers' variety-seeking behavior affect the service level and pricing decisions of both the traditional and sharing products? This study focuses on analyzing the experience of sharing products, so we concentrated on the services sharing rather than the goods sharing; this is because the SE provides for the diversified needs of consumers. Consumers are willing to accept services through different experiences and engage in variety-seeking behavior. We developed a simple model to explain why some customers choose traditional products while other customers choose sharing products, and we examined the case of room sharing. However, the model applies more broadly to other sharing products. We researched traditional service providers and sharing service providers, and both providers have a common target customer group. Our model is a two-period duopoly model with service level decisions and pricing strategies, in which two kinds of products compete for consumers in a Hotelling fashion. We derived the equilibrium outcome of their service level decisions as well as their pricing strategies. In this paper, there are two competitors: A traditional firm that provides normal or traditional products, denoted by the letter $N$, and an owner providing sharing and personalized products via a third-party sharing platform, denoted $S$. There are two types of consumers in our model: The first is a regular consumer whose purchasing decisions are only determined by their preferences and the products' prices, whereas the other is the variety-seeking consumer who incurs staying costs and gets bored through repeated purchases [12].

Our analysis revealed that, without considering the consumers' variety-seeking behavior, the traditional product attracted consumers with a high level of service and high price, while the sharing product attracted consumers with a low level of service and low price. When we only considered variety-seeking behavior and did not adjust the service level, the product with the low level of service benefited from the consumers' variety-seeking behavior, while the product with the high level of service lost profits. When we considered the variety-seeking behavior and adjusted the service level as well as the price, the sharing product was attractive to variety-seeking consumers and it gained a greater competitive advantage over the traditional product. For two periods, the number of variety-seeking consumers that switched from buying traditional products to buying sharing products was greater than those that switched from buying sharing products to buying traditional products. Furthermore, we found that when the consumers' variety-seeking behavior was not obvious, the number of consumers shifting from the traditional product increased monotonically. However, when the variety-seeking behavior was obvious, the number of consumers shifting from the traditional product decreased monotonically.

The rest of this paper is organized as follows. In Section 2, we present a literature review focusing on service sharing, variety-seeking behavior, and pricing strategies of the SE. In Section 3, we present our assumptions and explain the decision model. In Section 4, we analyze the pricing and service level decisions for two scenarios—without and with variety-seeking behavior. Our discussion and conclusions are presented in the last section. All of the proofs are given in the Appendix A.

## 2. Literature Review

In this section, we give a brief review of the literature pertinent to our research, namely, service sharing, variety-seeking behavior, and the pricing strategies of the SE.

### 2.1. Service Sharing

The SE emerged and disrupted established industries. In the service industry, researchers investigating service sharing mainly examined Airbnb and Uber via case studies. As are typical of disruptive innovations, Airbnb and Uber services were often cheaper than those of the traditional industries [2,13]. Moreover, Airbnb accommodations might provide for a more unique and authentic experience, and might offer useful household benefits (e.g., a kitchen) not typically available in hotels [13]. In particular, in the tourism industry, service sharing has played a positive role in promotion. Many studies have explained this positive phenomenon. Tussyadiah and Pesonen (2015) found that peer-to-peer accommodation significantly drove travelers to travel more often, stay longer, and participate in more activities [4]. Guttentag et al. (2018) identified five motivating factors—Interaction, Home Benefits, Novelty, Sharing Economy Ethos, and Local Authenticity—to use Airbnb instead of hotels [5]. Lin et al. (2019) identified three types of social contact in the Airbnb accommodation: Guest-host, guest-community, and guest-guest contacts [6]. In contrast, some scholars regarded distrust as the only factor constraining consumers from choosing Airbnb [14]. The above-mentioned studies show that the SE enables consumers to switch between the traditional and sharing products by some intrinsic factors.

### 2.2. Variety-Seeking Behavior

In this paper, we argue that variety-seeking behavior is driven by intrinsic motivations and occurs in a situation in which perceived differences among the alternatives are smaller [15]; that is, both traditional and sharing products have a common target customer group. We do not discuss the other situation; namely, that more variety in a product line leads to consumers' variety-seeking strategies [16]. Variety-seeking behavior has been studied in tourism literature [17–19]. Our paper is closely related to the following representative papers on variety-seeking behavior. Seetharaman and Che (2009) investigated the price competition between firms in a variety-seeking market by using a two-period duopoly framework and found that prices in both periods were higher than those in an otherwise identical market without variety seeking [9]. Sajeesh and Raju (2010) divided consumers into variety-seeking or not, and they studied competitive positioning and pricing strategies in markets in which consumers seek variety [20]. Unlike Sajeesh and Raju (2010), Niu, et al. (2019) examined jointed pricing and quality decisions considering customers' variety-seeking behavior. They found that the existence of variety-seeking customers reduced firms' incentives to improve quality levels [21]. Our research is different from the above works in that we involved the service level decision with the pricing strategy between traditional products and sharing products.

### 2.3. SE Pricing Strategy

Pricing is perhaps the most promising and important area for additional research on the SE [22]. Zervas et al. (2017) found that hotels affected by Airbnb have responded by reducing prices to attract consumers, not just participants in the sharing economy [7]. Fang et al. (2017) introduced a two-sided market model and compared pricing strategies that maximized revenue and maximized social welfare [23]. Kung and Zhong (2017) formulated a two-sided platform's profit maximization problem and analyzed three pricing strategies (membership-based pricing, transaction-based pricing, and cross-subsidization) [24]. However, the latter two studies discussed the platform's price strategies, but they did not consider the competition between the sharing market and traditional market. Fraiberger and Sundararajan (2015) developed a new dynamic model of peer-to-peer Internet-enabled rental markets for durable goods in which consumers were heterogeneous in their price sensitivity and asset

utilization rates [25]. Jiang and Tian (2018) studied the traditional firm's optimal pricing strategy when faced with consumers who could share products with other consumers through sharing platforms [26]. Tian and Jiang (2018) also found that product sharing affected the distribution channel [27]. Liu, et al. (2019) examined the impact of the SE on a manufacturer's product rollover strategy [28]. Feng, et al. (2019) also discussed a manufacturer's optimal business strategy involving the interaction of product sharing and product rollover in the SE [29]. The above-mentioned studies assumed that the two competing products were homogeneous and that consumers had no obvious preference for purchasing certain products. Narasimhan et al. (2016) presented important questions with implications for both SE providers and legacy providers on how they should compete. Their findings suggested that both should not focus exclusively on price for a competitive advantage, but should also compete on service features [30]. The above studies provided new ideas for this research's joint decisions about the pricing and service level.

## 3. Model

We mainly focused on the service sharing of products that consumers repeatedly buy. We assumed that there is a traditional firm that provides a traditional product and an owner who rents out a sharing product via a third-party sharing platform. There are two kinds of products that are heterogeneous but functionally substitutable (e.g., Uber/Airbnb versus Taxi/Hilton), and both products have a common target customer group. Similarly to other studies [31–33], we modeled the consumer's preferences along the Hotelling line [34,35]. The consumer is represented by $\theta$ on the unity line. Each consumer's ideal product $\theta \in [0, 1]$ is characterized by the consumer's location on the unit interval. Similarly to Hotelling-like competition models, the traditional firm and owner each have a base product, and those products are located at the opposite ends of the unit interval [32,35]. The distance between a consumer and the firm/owner's product position results in disutility or the consumer's sacrifice relative to the consumer's preferred product configuration [36].

In addition to the basic functions, both the traditional product and sharing product have characteristics $f_i$ ($i = N, S$) that may be a set of preferences or the service level of the product, such as the standardized service level of Taxi/Hilton and the personalization level of Uber/Airbnb. The standardized service level is represented by $e$, and the personalized service level is represented by $\omega$. In the current market, the personalization attribute of a product manifests itself as a pivotal factor in the consumer's purchasing decision. The term "personalization" in our model coincides with the difference between the sharing service and standardized service. Personalization includes customized solutions, convenience, refreshment, involvement, interpersonal relationships, and local culture [37,38]. However, these personalized factors must be recognized by consumers. In other words, consumers who perceive sustainability and greenness are more willing to participate in the SE [37]. The consumer's recognition degree of a standardized service and a personalized service are represented by $\alpha$ ($\alpha \in [0, 1]$) and $\beta$ ($\beta \in [0, 1]$), respectively. We used a study from the literature [33] to propose the following utility function to measure the consumer's positive utility of the standardized service and the personalized service of the product:

$$f_i(e, \omega) = \alpha e + \beta \omega. \tag{1}$$

Therefore, the utility of consumer $\theta$ or product $i$ with $e$ and $\omega$ is defined by:

$$U_i(p_i|\eta_i, \theta) = u - d_i(\eta_i - \theta)^2 + f_i(e, \omega) - p_i \tag{2}$$

where $u$ represents the consumer's basic utility. For example, the accommodation function of a hotel can meet the consumer's basic utility. $\eta_i - \theta$ quantifies the distance between the consumer's ideal product and the available ones, $\eta_N = 0$ and $\eta_S = 1$. The intensity of the consumer's preference for product $i$ is denoted $d_i$. Therefore, we express the disutility of the misfit due to the customer's sacrifice from their ideal product as $d_i(\eta_i - \theta)^2$. $f_i(e, \omega)$ measures the positive utility due to the standardized service and the personalized service of the product, particularly $f_N(e, \omega) = \alpha e$ and $f_S(e, \omega) = \beta \omega$. $p_i$ is

the price of product *i*. The price will affect the consumer's decision of whether to buy the product and which product to buy among the available alternatives. The consumer's individual decisions are largely based on utility-maximizing behavior.

We assumed that the product cost is quadratic with the service level offered to consumers [39–41]. The cost of product *N* can be expressed as $k_1 e^2$, where $k_1$ is the cost coefficient of a standardized service. The cost of product *S* mainly includes the owner's moral hazard cost that is related to the personalized service level of product *S* [26]. The cost of product *S* is denoted $k_2 \omega^2$, where $k_2$ is the cost coefficient that measures the moral hazard cost. To facilitate analysis, we normalize $k_1 = k_2 = k$. For the owner of the sharing platform, if the owner rents out the product *S*, they will earn the rental fee but need to pay the sharing platform a percentage fee, denoted $\lambda \in [0, 1]$, a fraction of the rental fee. Typically, in practice, the sharing platform collects the rental fee from the consumer, keeps a fixed $\lambda$ fraction of that fee as a service charge, and will give the remaining fraction $(1 - \lambda)$ to the owner [26]. For the sharing platform, there is a certain operation and management cost which has no effect on the conclusions of this paper. Without a loss of generality, we normalize the cost to 0.

There are two types of consumers in the market: Regular consumers and variety-seeking consumers. To express the two types of consumers' periodic purchasing behavior, we assumed that there are two sales periods, *j* = 1, 2. During the first period, consumers make purchasing decisions based on their true valuations and the products' prices. During the second period, regular consumers still purchase the product depending on their preferences and the products' prices, whereas the variety-seeking consumers will incur staying costs and become bored with repeatedly making purchases. We denote the consumer's boredom sensitivity by $\gamma$.

The timing of events in the core model is as shown in Figure 1. First, in the beginning of period 1, the traditional firm and the owner simultaneously decide their products' service levels *e* and $\omega$. Second, the traditional firm and the owner simultaneously decide their products' prices based on the service levels. Third, consumers make their purchasing decisions. The decision sequence of period 2 is the same as that of the period 1.

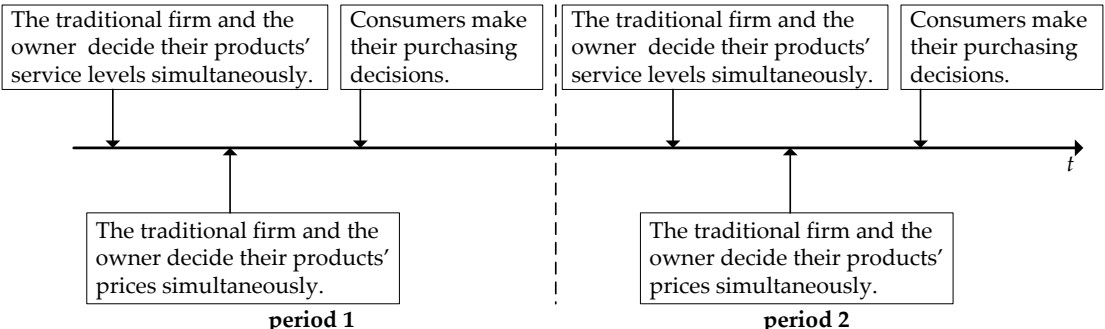

**Figure 1.** Decision sequence.

The notations used in this paper are shown in Table 1.

**Table 1.** Variables and definitions used in the model.

| Variables | Definitions |
|---|---|
| $i$ | Product, $i = N$ (traditional product), $S$ (sharing product). |
| $j$ | Period, $j = 1, 2$. |
| $U_{i,j}$ | Consumer's utility of choosing product $i$ at period $j$. |
| $\theta$ | Consumer type, $\theta \in [0, 1]$. |
| $\theta_m$ | Boundary consumer who is indifferent to buying either of the two kinds of products, $m \in \{0, 1, 2\}$. |
| $\eta_i$ | Product's positions. |
| $e_j$ | Standardized service level of traditional product in period $j$. |
| $\omega_j$ | Personalized service level of sharing product in period $j$. |
| $\alpha$ | Consumer's recognition degree of standardized service, $\alpha > 0$. |
| $\beta$ | Consumer's recognition degree of personalized service, $\beta > 0$. |
| $u$ | Consumer's intrinsic utility. |
| $d_i$ | Intensity of the consumer's preference for product $i$. |
| $\delta$ | Fraction of the variety-seeking customers in the market, $\delta \in [0, 1]$. |
| $p_{i,j}$ | Price of product $i$ in period $j$. |
| $q_{i,j}$ | Demand for product $i$ in period $j$. |
| $k$ | Constant, $k > 0$. |
| $\lambda$ | Sharing platform's percentage fee, $\lambda \in [0,1)$. |
| $\gamma$ | Consumer's boredom sensitivity, $\gamma \in [0,1)$. |
| $\pi_{N,j}$ | Profits of traditional firm $N$ in period $j$. |
| $\pi_{SP,j}$ | Profits of sharing platform $S$ in period $j$. |
| $\pi_{SO,j}$ | Earnings of owner in period $j$. |
| $NV$ | Benchmark with no variety-seeking behavior, denoted by superscript $NV$. |
| $V$ | Variety-seeking behavior, denoted as a superscript $V$. |

## 4. Analysis

In this section, we analyze two scenarios. The first scenario is a benchmark case in which consumers do not exhibit variety-seeking behavior during the two periods. This scenario is hereafter denoted by the superscript $NV$. In the second scenario, consumers exhibit variety-seeking behavior during the two periods. Variety-seeking consumers become bored with the purchased products and are more willing to choose products that they have not bought before. This scenario is denoted by the superscript $V$.

### 4.1. Benchmark Case: Without Variety Seeking

We first analyze a benchmark case in which consumers do not exhibit variety-seeking behavior during the two periods; that is, all of the consumers in the market are regular consumers whose purchasing decisions are only determined by their preferences and the products' prices. In this case, the consumer's utility functions are as follows. The consumer's utility is $U_{N,j}^{NV}(\theta|e) = u - d\theta^2 + \alpha e_j - p_{N,j}$ for buying the traditional product and $U_{S,j}^{NV}(\theta|\omega) = u - d(1 - \theta)^2 + \beta\omega_j - p_{S,j}$ for buying the sharing product, where $j = 1, 2$. The following lemma characterizes the equilibrium solutions under the pure strategy equilibria. All of the proofs are presented in the Appendix A.

**Lemma 1.** *Without considering the consumer's variety-seeking behavior, we have $e > \omega$ and $p_N > p_S$ for each period.*

Lemma 1 indicates that the traditional product attracts consumers with its high level of service and high price, while the sharing product attracts consumers with a low level of service and low price. In reality, consumers always rent the sharing product from the owner via the third-party sharing platform. The owner pays the sharing platform a certain rental fee that is an extra charge not included in the price of the traditional product. Therefore, to reduce the cost, we find that the service level of the sharing product is lower than that of the traditional product, as shown in Figure 2a. Figure 2a also shows that the sharing product's service level is positively correlated with the consumer's recognition degree of

personalized service. That is to say, the owner will have an incentive to improve the sharing product's personalized service if consumers are satisfied with the personalized service. Figure 2a shows that whether $\beta$ is large or small will not significantly change the traditional product's standardized service. Furthermore, the price of the sharing product is lower than that of the traditional product, as shown in Figure 2b. The sharing product's price is positively correlated with the consumer's recognition degree of personalized service. The traditional product's price will be reduced slightly to attract consumers who switch to the sharing product. In summary, without considering the consumer's variety-seeking behavior, the traditional product attracts consumers via its high level of service and high price, while the sharing product attracts consumers through its low level of service and low price.

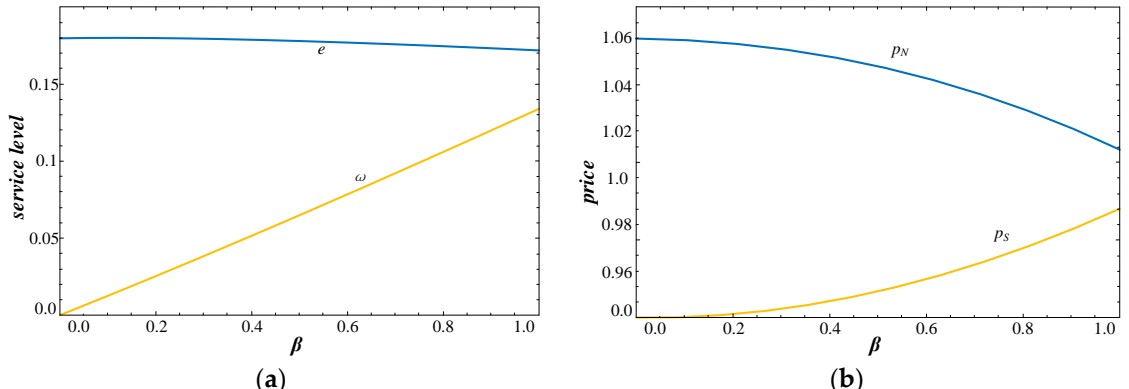

**Figure 2.** Service levels and prices of two products without variety seeking ($\alpha = 1$). Note: (**a**) Service levels; (**b**) prices.

### 4.2. Variety-Seeking Behavior

In this subsection, we analyze a case where consumers become bored when making their purchasing decisions at the second period. Note that regular consumers will not become bored, since their ex ante purchase decisions are based on their true valuations. By contrast, variety-seeking consumers will incur staying costs and become bored through repeated purchases [12]. They will reduce their preference for the products they bought before. However, it does not change the valuation of other unpurchased products. For example, if a variety-seeking consumer has experienced products with standardized services, they will become bored with the experienced products when they choose to buy similar products again.

To model such boredom, we assumed that variety-seeking consumers have an aversion to products that have been purchased during the last period. Accordingly, the variety-seeking consumers' valuation of purchased products decays over time at $\gamma e$ or $\gamma \omega$, where $\gamma$ is the consumer's boredom sensitivity. To ensure the positive utility of $f_i(e,\omega)$, let $0 \leq \gamma \leq \min\{\alpha, \beta\}$. Therefore, the variety-seeking consumers' net utilities are as follows:

(1)    Purchase product N during the first period and purchase product N during the second period.

$$U_{N-N,2}^{V}(\theta|e) = u - d\theta^2 + (\alpha - \gamma)e_2 - p_{N,2} \tag{3}$$

(2)    Purchase product S during the first period and purchase product S during the second period.

$$U_{S-S,2}^{V}(\theta|\omega) = u - d(1 - \theta_2)^2 + (\beta - \gamma)\omega_2 - p_{S,2} \tag{4}$$

The fraction of variety-seeking customers in the market is denoted $\delta$, and $0 \leq \delta \leq 1$. Consumers with the remaining $1 - \delta$ fraction are regular consumers. Note that the benchmark is a special case where $\delta = 0$ or $\gamma = 0$. Next, we analyze the case in which $\delta \in (0, 1]$ and $\gamma > 0$. In addition, we analyze

the impact of variety-seeking behavior on providers' strategies, which are divided into two scenarios based on the providers' responses. One scenario is that the product's provider only adjusts the price without adjusting the service level. The other scenario is that the provider adjusts both the product's price and service level.

### 4.2.1. No Adjustment to the Service Level

**Lemma 2.** *When two products' service levels remain unchanged for two periods, whether the product's price decreases or increases is related to e and $\omega$. (i) If $e = \omega$, the two kinds of products' prices remain unchanged for two periods. (ii) If $e > \omega$, product N's price decreases and product S's price increases. (iii) If $e < \omega$, product N's price increases and product S's price decreases. (iv) The extent of the changes in the product's price is related to the fraction of variety-seeking customers ($\delta$) and the consumers' boredom sensitivity ($\gamma$). Mathematically, $\Delta p_N = p_{N,2} - p_{N,1} = \frac{1}{3}\delta \gamma(-e + \omega)$, and $\Delta p_S = p_{S,2} - p_{S,1} = \frac{1}{3}\delta \gamma(e - \omega)$.*

For the proof of Lemma 2, see the Appendix A.

Lemma 2 shows that two competing products' prices remain unchanged when consumers feel that there is no difference between the two products' service levels (i.e., $e = \omega$). However, if they feel there is a difference, the product with a high service level will be reduced in price to cater to the variety-seeking customers. This is because once a variety-seeking customer has experienced a product's high service during the first period, their marginal utility of the same service will decrease because of the staying cost during the second period. When the market inevitably loses customers, the product's provider will lower the price to attract customers. Consequently, the product's profit decreases. However, the product with a low service level gains a competitive advantage with its lower price during the first period. During the second period, the product's price and profit increase. This occurs because more variety-seeking customers shift from the other products. As a result, the product's price and profit will increase accordingly. The product with the low service level will benefit from the consumers' variety-seeking behavior.

Furthermore, in Lemma 2, the fraction of variety-seeking customers ($\delta$) and the consumers' boredom sensitivity ($\gamma$) do not affect the increase or decrease of the product's price. However, they affect the extent of changes in the product's price. A high fraction of variety-seeking customers or high consumers' boredom sensitivity will result in a larger price adjustment.

### 4.2.2. Adjusting Service Level

From Lemma 2, we learned that the provider only adjusts the product's price when there are variety-seeking customers in the market. However, for the provider with a high level of service, only adjusting the price is not its dominant strategy. Since the product's provider with the high service level loses profits, the low-level service provider benefits from the variety-seeking behavior. As a consequence, the product's provider will respond by adjusting the service level as well as the price.

We can now compare the equilibrium solutions without variety-seeking behavior and with variety-seeking behavior. We obtain Propositions 1–3.

**Proposition 1.** *Considering the consumer's variety-seeking behavior, we have $e_2^V < e_2^{NV} = e_1^{NV} = e_1^V$, $\omega_2^V < \omega_2^{NV} = \omega_1^{NV} = \omega_1^V$, $p_{N,2}^{NV} > p_{N,2}^V > p_{S,2}^V > p_{S,2}^{NV}$, $\pi_{SO,2}^V > \pi_{SO,2}^{NV}$, $\pi_{N,2}^V > \pi_{N,2}^{NV}$.*

**Proposition 2.** *In this equilibrium, as the consumer's boredom sensitivity ($\gamma$) increases, the increase in the sharing product's profits is greater than that of the traditional product's profits; mathematically, $\frac{\partial \pi_{N,2}^V}{\partial \gamma} < \frac{\partial \pi_{SO,2}^V}{\partial \gamma}$. As $\gamma$ increases, the decrease in the traditional product's service level is greater than in the sharing product's service level; mathematically, $\left|\frac{\partial e_2^V}{\partial \gamma}\right| > \left|\frac{\partial \omega_2^V}{\partial \gamma}\right|$.*

**Proposition 3.** *The number of variety-seeking consumers who switch from buying traditional products to buying sharing products is greater than the number who switch from buying sharing products to buying traditional products.*

According to Lemma 2 and Proposition 1, the traditional product's price decreases and the sharing product's price increases regardless of the service level, since, in essence, variety-seeking consumers experience the differentiated product rather than the product with a high level of service. Thus, the traditional product's high level of service and high price will drive away consumers who are only price sensitive but not variety seeking. The traditional product's service level and price are only reduced to cut costs and attract consumers during the second period.

The sharing product with the personalized service is attractive to variety-seeking consumers and it gains a greater competitive advantage. Although the traditional product's provider lowers the service level and price to win back consumers, the sharing product is still set at a higher price to benefit more from its competitive advantage. In addition, the owner will also lower the personalized service level of the sharing product to cut costs. Therefore, the increase in price, decrease in the service level, and expansion in the market demand cause the profits of the sharing product to increase, while the decrease in the service level and reduction in cost still make the traditional product benefit more than before. The results are as shown in Proposition 2. The consumer's variety-seeking behavior has a larger impact on the profit of the sharing product than that of the traditional product (i.e., $\frac{\partial \pi_{N,2}^V}{\partial \gamma} < \frac{\partial \pi_{SO,2}^V}{\partial \gamma}$), while the decrease in the service level of the traditional product will be less sensitive to the variety-seeking behavior than that of the sharing product (i.e., $\left|\frac{\partial e_2^V}{\partial \gamma}\right| > \left|\frac{\partial \omega_2^V}{\partial \gamma}\right|$).

Proposition 3 is a comparison of the number of consumers who variably seek different products. As shown in Figure 3, the number of variety-seeking consumers who switch from buying traditional products to buying sharing products is greater than the number who switch from buying sharing products to buying traditional products. There are two reasons for this. One is that lowering the service level of the traditional product will drive consumers who preferred standardized service before to the sharing product. Second, consumers who use sharing products tend to have variety-seeking behavior. Their preference for new things will reduce their interest in traditional products. Therefore, variety-seeking behavior has a greater impact on the loss of customers for the traditional product than the sharing product.

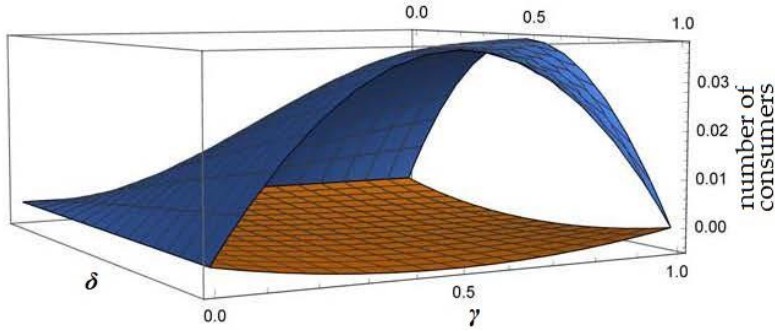

**Figure 3.** Number of consumers who variably seek different products ($k = 1$, $\alpha = \beta = 1$, $\lambda = 0.2$).

In Figure 3, as the fraction of variety-seeking customers ($\delta$) and the consumers' boredom sensitivity ($\gamma$) increase simultaneously, the impact of $\delta$ and $\gamma$ on the number of shifting consumers who switch from buying traditional products to buying sharing products is non-monotonic. We find that consumers' variety-seeking behavior is not obvious (i.e., $\delta \in (0, 0.5]$ and $\gamma \in (0, 0.5]$), and the number of consumers shifting from the traditional product increases monotonically with $\delta$ and $\gamma$. While variety-seeking behavior is obvious (i.e., $\delta \in (0.5, 1]$ and $\gamma \in (0.5, 1]$), the number of consumers shifting from the traditional product decreases monotonically with $\delta$ and $\gamma$. When consumers' variety-seeking behavior

is not obvious (i.e., $\gamma \in (0, 0.5]$), the two kinds of products' service levels and prices do not change significantly, as shown in Figure 4. Thus, more consumers move away from the traditional product with a high price and high level of service than do from the sharing product. However, when consumers' variety-seeking behavior is obvious (i.e., $\gamma \in (0.5,1]$), the prices of both products are adjusted dramatically to be closer to each other, as shown in Figure 4a. The closer prices of both products reduce the variety-seeking consumers' sensitivity to price and mitigate the consumers' loss on the traditional product.

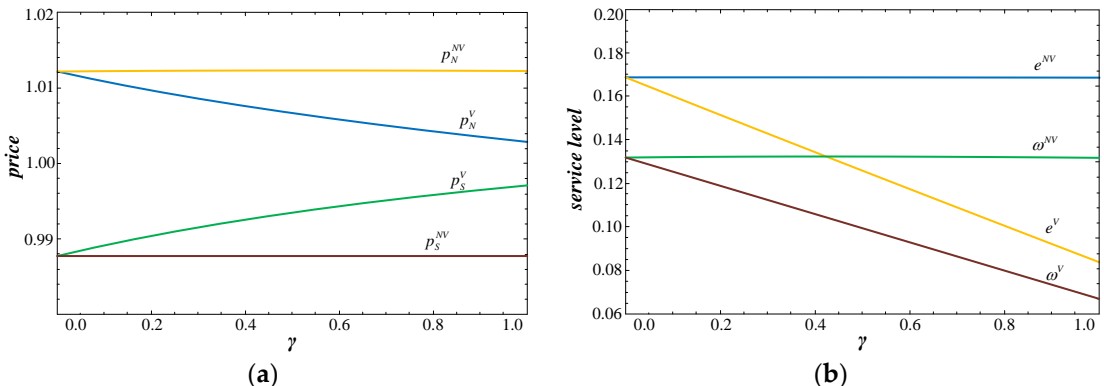

**Figure 4.** Prices and service levels of two products for two periods ($\delta$ = 0.5). Note: (**a**) Prices; (**b**) service levels.

Owing to the above analysis, considering the consumers' variety-seeking behavior, the product's provider has fewer incentives to maintain and improve the product's service level. In particular, the more variety-seeking consumers there are in the market, the lower the willingness of the product's provider to improve the service level. In contrast, the providers will reduce the service level of their own products to narrow the difference with competitors' products. This will lead to the product differentiation in the market being small. Thus, the competitive strategy of the product's providers will transform from product differentiation to price reduction, which is not conducive to market development. Furthermore, our analysis results are in accord with those of Guttentag et al. (2018) [5]. What attracts consumers most to a sharing product is low price or cost, and services such as interaction and novelty are generally secondary.

Therefore, a product's provider should focus more on weakening consumer's variety-seeking behavior and reducing consumers' staying cost. Taking Haidilao as an example in Hot Pot catering, it always provides innovative and pleasant services to their consumers. These services include special personnel watching over children, free manicures for female customers, free leather shoe care, free photo printing, Sichuan-featured face-changing shows, and making noodles combined with Chinese martial arts. Their high-quality services have made Haidilao a world-renowned catering brand after over 20 years of development.

## 5. Discussion and Conclusions

An SE increases product diversity and provides variety-seeking consumers with more purchasing options than traditional products. In this research, we provided a simple model to demonstrate how consumers' variety-seeking behavior affected the service level and pricing decisions of both traditional and sharing products. We created a two-period duopoly model with service level and pricing decisions in which two kinds of products competed for consumers in a Hotelling fashion.

First, without considering the consumers' variety-seeking behavior, our analysis revealed that traditional products attract consumers with a high level of service and high price, while sharing products attracted consumers with a low level of service and low price. Second, by considering variety-seeking behavior and without adjusting the service level, we learned that the product with the

low level of service would benefit from the consumers' variety-seeking behavior, while the product with the high level of service would lose profits. The higher the fraction of variety-seeking customers or the higher the consumers' boredom sensitivity, the larger the extent of the price adjustment. However, for the provider with a high level of service, only adjusting the price is not its dominant strategy. Third, by considering variety-seeking behavior and adjusting the service level as well as the price, the sharing product was attractive to variety-seeking consumers and gained a greater competitive advantage than that of the traditional product. For two periods, the number of variety-seeking consumers who switched from buying traditional products to buying sharing products was greater than the number who switched from buying sharing products to buying traditional products. Furthermore, we found that when the consumers' variety-seeking behavior was not obvious, the number of consumers shifting from the traditional product increased monotonically. When the variety-seeking behavior was obvious, the number of consumers shifting from the traditional product decreased monotonically.

## 6. Limitations and Future Research

We have simplified our model specifications to derive meaningful analytical results. Some of our assumptions can be relaxed to deal with more complex situations. First, we assumed that there was only one traditional firm and one owner in the market. The competition among more traditional firms and owners should be taken into account in the future. Second, we assumed that the traditional firm only provided the traditional product through offline channels. The situation in which the traditional firm joins the SE to provide both the sharing product and traditional product should be considered in the future.

**Author Contributions:** Conceptualization, Z.L. and J.F.; writing—original draft preparation, Z.L.; writing—review and editing, Z.L. and J.F.; project administration, B.L.

**Funding:** This research was funded by National Natural and Science Foundation of China with grant number 71971134; Innovation Method Fund of China with grant number 2018IM020300; Industrial and Informationalization Ministry of China for Cruise Program with grant number 2018-473; and Humanities and Social Sciences Foundation of Ministry of Education in China with grant number 17YJC630091.

**Conflicts of Interest:** The authors declare that there is no conflict of interest regarding the publication of this paper.

## Appendix A

This appendix contains all of the proofs of the lemmas and propositions in the paper.

**Proof of Lemma 1.** We obtain feedback equilibrium solutions using backwards induction in the first scenario.

At period 1, we solve $U_{N,1}^{NV}(\theta|e) = U_{S,1}^{NV}(\theta|\omega)$, and derive the regular consumer's indifferent point, as shown in Figure A1:

$$\theta_0 = \frac{1}{2}\left(1 + \alpha e_1 - \beta\omega_1 - p_{N,1} + p_{S,1}\right) \tag{A1}$$

Then, the demands of two products are:

$$q_{N,1}^{NV} = \theta_0 = \frac{1}{2}\left(1 + \alpha e_1 - \beta\omega_1 - p_{N,1} + p_{S,1}\right) \tag{A2}$$

$$q_{S,1}^{NV} = 1 - \theta_0 = \frac{1}{2}\left(1 - \alpha e_1 + \beta\omega_1 + p_{N,1} - p_{S,1}\right). \tag{A3}$$

Therefore, the traditional firm's profits and the owner's earnings are:

$$\pi_{N,1}^{NV} = p_{N,1} \cdot q_{N,1}^{NV} - ke_1^2 = \frac{1}{2}p_{N,1}\left(1 + \alpha e_1 - \beta\omega_1 - p_{N,1} + p_{S,1}\right) - ke_1^2 \tag{A4}$$

$$\pi_{SO,1}^{NV} = p_{S,1} \cdot q_{S,1}^{NV} - k\omega_1^2 = \frac{1}{2}p_{S,1}\left(1 - \alpha e_1 + \beta\omega_1 + p_{N,1} - p_{S,1}\right) - k\omega_1^2. \tag{A5}$$

According to $H(p_{N,1}, p_{S,1}) < 0$, there exist optimal solutions, i.e., $\frac{\partial \pi_{N,1}}{\partial p_{N,1}} = 0$ and $\frac{\partial \pi_{SO,1}}{\partial p_{S,1}} = 0$. We get:

$$p_{N,1} = \frac{1}{3}(3 + \alpha e_1 - \beta \omega_1) \tag{A6}$$

$$p_{S,1} = \frac{1}{3}(3 - \alpha e_1 + \beta \omega_1). \tag{A7}$$

Substituting (A6) and (A7) into Equations (A4) and (A5), we get:

$$\pi_{N,1}^{NV} = \frac{1}{18}\Big[ e_1^2\big(-18k + \alpha^2\big) - 2\alpha e_1 + (\beta \omega_1 - 3)^2 \Big] \tag{A8}$$

$$\pi_{SO,1}^{NV} = \frac{1}{18}(1 - \lambda)(3 - e_1 + \beta \omega_1)^2 - k\omega_1^2. \tag{A9}$$

According to $H(e_1, \omega_1) < 0$, let $\frac{\partial \pi_{N,1}}{\partial e_1} = 0$, $\frac{\partial \pi_{SO,1}}{\partial \omega_1} = 0$. We obtain:

$$e_1 = \frac{9k\alpha - \alpha\beta^2 + \alpha\beta^2\lambda}{3k(18k - \alpha^2 - \beta^2 + \beta^2\lambda)} \tag{A10}$$

$$\omega_1 = \frac{9k - \alpha^2 - 9k\lambda + \alpha^2\lambda}{3k(18k - \alpha^2 - \beta^2 + \beta^2\lambda)}. \tag{A11}$$

Substituting Equations (A10) and (A11) into Equations (A6)–(A9), we get the equilibrium solutions without variety seeking as follows:

(1)   The service levels are $e_1^{NV} = e_2^{NV} = \frac{\alpha\left(9k - \beta^2 + \beta^2\lambda\right)}{3k(18k - \alpha^2 - \beta^2 + \beta^2\lambda)}$, $\omega_1^{NV} = \omega_2^{NV} = \frac{\beta\left(9k - \alpha^2 - 9k\lambda + \alpha^2\lambda\right)}{3k(18k - \alpha^2 - \beta^2 + \beta^2\lambda)}$.

(2)   The prices are $p_{N,1}^{NV} = p_{N,2}^{NV} = \frac{2\left(9k + \beta^2 - \beta^2\lambda\right)}{18k - \alpha^2 - \beta^2 + \beta^2\lambda}$, $p_{S,1}^{NV} = p_{S,1}^{NV} = \frac{2\left(9k - \alpha^2\right)}{18k - \alpha^2 - \beta^2 + \beta^2\lambda}$.

(3)   The profits are $\pi_{N,1}^{NV} = \pi_{N,2}^{NV} = \frac{\left(18k - \alpha^2\right)\left(9k + \beta^2 - \beta^2\lambda\right)^2}{9k(18k - \alpha^2 - \beta^2 + \beta^2\lambda)^2}$, $\pi_{SO,1}^{NV} = \pi_{SO,1}^{NV} = \frac{(1-\lambda)\left(9k - \alpha^2\right)^2\left(18k - \beta^2 + \beta^2\lambda\right)^2}{9k(18k - \alpha^2 - \beta^2 + \beta^2\lambda)^2}$.

At period 2, the regular consumer's purchasing decision has not changed, so the equilibrium solutions are the same as those at period 1.

According to the equilibrium solutions of two periods, we obtain

$$e_1^{NV} - \omega_1^{NV} = \frac{9k(\alpha + \beta(-1 + \lambda)) - \alpha\beta(\alpha - \beta)(-1 + \lambda)}{3k(18k - \alpha^2 + \beta^2(-1 + \lambda))} > 0$$

$$p_{N,1}^{NV} - p_{S,1}^{NV} = \frac{2\left(\alpha^2 + \beta^2(-1 + \lambda)\right)}{18k - \alpha^2 + \beta^2(-1 + \lambda)} > 0$$

$\square$

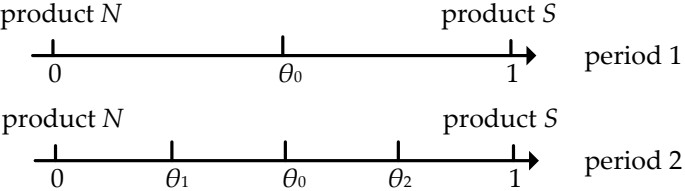

**Figure A1.** Regular consumer's utilities over two periods.

**Proof of Lemma 2.** At period 1, when there are both regular consumers and variety-seeking consumers in the market, the prices, service levels, and profits of two products are same as the benchmark at the first period.

At period 2, the regular consumer's purchasing decision has not changed. However, the variety-seeking consumer's utilities are as follows.

If the variety-seeking consumer purchases product $N$, then they continue to purchase product $N$. Then they will derive $U^V_{N-N,2}(\theta|e) = u - d\theta^2 + (\alpha - \gamma)e_2 - p_{N,2}$. However, they switch to purchase product $S$, and then they will derive $U^V_{N-S,j}(\theta|\omega) = u - d(1-\theta)^2 + \beta\omega_2 - p_{S,2}$. Therefore, the variety-seeking consumer's indifferent point is $\theta_1 = \frac{1+(\alpha-\gamma)e_2-p_{N,2}-\beta\omega_2-p_{S,2}}{2}$, as shown in Figure A2.

If the variety-seeking consumer purchases product $S$, then they continue to buy product $S$, and will derive $U^V_{S-S,2}(\theta|\omega) = u - d(1-\theta)^2 + (\beta - \gamma)\omega_2 - p_{S,2}$. However, if they switch to purchase product $N$, then they will derive $U^V_{S-N,j}(\theta|e) = u - d(1-\theta)^2 + \beta e_2 - p_{N,2}$, and the indifferent point is $\theta_2 = \frac{1-(\beta-\gamma)\omega_2+p_{S,2}-\alpha e_2-p_{N,2}}{2}$, as shown in Figure A2.

There is a fraction ($\delta$) of the variety-seeking consumers in the market and a fraction $(1-\delta)$ of the regular consumers. Thus, the demand of product $N$ during the second period is:

$$q^V_{N,2} = (1-\delta)\theta_0 + \delta\theta_1 + \delta(\theta_0 - \theta_1) = \frac{1 + (\alpha - \delta\gamma)e_2 - (\beta - \delta\gamma)\omega_2 - p_{N,2} + p_{S,2}}{2} \quad \text{(A12)}$$

$$q^V_{S,2} = (1-\delta)(1-\theta_0) + \delta(1-\theta_2) + \delta(\theta_2 - \theta_0) = \frac{1 - (\alpha - \delta\gamma)e_2 + (\beta - \delta\gamma)\omega_2 + p_{N,2} - p_{S,2}}{2}. \quad \text{(A13)}$$

Then, the profits of product $N$ are:

$$\pi^V_{N,2} = p_{N,2}\cdot q_{N,2} - ke_2^2 = \frac{p_{N,2}\left[1 + (\alpha - \delta\gamma)e_2 - (\beta - \delta\gamma)\omega_2 - p_{N,2} + p_{S,2}\right]}{2} - ke_2^2 \quad \text{(A14)}$$

and the profits of product $S$ are:

$$\pi^V_{SO,2} = p_{S,2}\cdot q_{S,2} - k\omega_2^2 = \frac{(1-\lambda)p_{S,2}\left[1 - (\alpha - \delta\gamma)e_2 + (\beta - \delta\gamma)\omega_2 + p_{N,2} - p_{S,2}\right]}{2} - k\omega_2^2. \quad \text{(A15)}$$

We use backward induction to solve this game.

According to $H(p_{N,2}, p_{S,2}) < 0$, we let $\frac{\partial \pi_{N,2}}{\partial p_{N,2}} = 0$ and $\frac{\partial \pi_{SO,2}}{\partial p_{S,2}} = 0$, we obtain:

$$p_{N,2} = \frac{3 + (\alpha - \delta\gamma)e_2 - (\beta - \delta\gamma)\omega_2}{3} \quad \text{(A16)}$$

$$p_{S,2} = \frac{3 - (\alpha - \delta\gamma)e_2 + (\beta - \delta\gamma)\omega_2}{3}. \quad \text{(A17)}$$

Substituting Equations (A16) and (A17) into Equations (A14) and (A15), we get:

$$\pi^V_{N,2} = \frac{1}{18}\left[e_2^2\left(-18k + (\alpha - \delta\gamma)^2\right) - 2(\alpha - \delta\gamma)e_2 + (3 - \beta(\alpha - \delta\gamma)\omega_2)^2\right] \quad \text{(A18)}$$

$$\pi^V_{SO,2} = \frac{1}{18}\left\{ \begin{array}{l} (1-\lambda)[3 - (\alpha - \delta\gamma)e_2]^2 \\ +2(1-\lambda)(\beta - \delta\gamma)[3 - (\alpha - \delta\gamma)e_2]\omega_2 \\ +\left[18k - (1-\lambda)(\beta - \delta\gamma)^2\right]\omega_2 \end{array} \right\}. \quad \text{(A19)}$$

According to $H(e_1, \omega_1) < 0$, and letting $\frac{\partial \pi_{N,2}}{\partial e_2} = 0$ and $\frac{\partial \pi_{SO,2}}{\partial \omega_2} = 0$, we get:

$$e_2 = \frac{(\alpha - \delta\gamma)\left[9k - (1-\lambda)(\beta - \delta\gamma)^2\right]}{3k\{18k - \alpha^2 - \beta^2 + 2\alpha\delta\gamma + (\beta - \delta\gamma)[\beta\lambda + \delta\gamma(2 - \lambda)]\}} \quad \text{(A20)}$$

$$\omega_2 = \frac{(\beta - \delta\gamma)\big[9k - (1-\lambda)(\alpha - \delta\gamma)^2\big]}{3k\{18k - \alpha^2 - \beta^2 + 2\alpha\delta\gamma + (\beta - \delta\gamma)[\beta\lambda + \delta\gamma(2-\lambda)]\}}. \tag{A21}$$

Substituting Equations (A20) and (A21) into Equations (A16)–(A19), we get the equilibrium solutions with variety seeking as follows:

(1) The service levels are $e_2^V = \frac{(\alpha-\delta\gamma)\big[9k-(1-\lambda)(\beta-\delta\gamma)^2\big]}{3k\{18k-\alpha^2-\beta^2+2\alpha\delta\gamma+(\beta-\delta\gamma)[\beta\lambda+\delta\gamma(2-\lambda)]\}}$, $\omega_2^V = \frac{(\beta-\delta\gamma)\big[9k-(1-\lambda)(\alpha-\delta\gamma)^2\big]}{3k\{18k-\alpha^2-\beta^2+2\alpha\delta\gamma+(\beta-\delta\gamma)[\beta\lambda+\delta\gamma(2-\lambda)]\}}$.

(2) The prices are $p_{N,2}^V = \frac{2\big[9k-(1-\lambda)(\beta-\delta\gamma)^2\big]}{18k-\alpha^2-\beta^2+2\alpha\delta\gamma+(\beta-\delta\gamma)[\beta\lambda+\delta\gamma(2-\lambda)]}$, $p_{S,2}^V = \frac{18k-2(\alpha-\delta\gamma)^2}{18k-\alpha^2-\beta^2+2\alpha\delta\gamma+(\beta-\delta\gamma)[\beta\lambda+\delta\gamma(2-\lambda)]}$.

(3) The profits are $\pi_{N,2}^V = \frac{\big[18k-(\alpha-\delta\gamma)^2\big]\big[9k-(1-\lambda)(\alpha-\delta\gamma)^2\big]^2}{9k\{18k-\alpha^2-\beta^2+2\alpha\delta\gamma+(\beta-\delta\gamma)[\beta\lambda+\delta\gamma(2-\lambda)]\}^2}$, $\pi_{SO,2}^V = \frac{(1-\lambda)\big[9k-(\alpha-\delta\gamma)^2\big]^2\big[18k-(1-\lambda)(\beta-\delta\gamma)^2\big]}{9k\{18k-\alpha^2-\beta^2+2\alpha\delta\gamma+(\beta-\delta\gamma)[\beta\lambda+\delta\gamma(2-\lambda)]\}^2}$.

Then, when two products' service levels remain unchanged for two periods, the difference between $p_{N,2}$ and $p_{N,1}$ is $\Delta p_N = p_{N,2} - p_{N,1} = \frac{3+(\alpha-\delta\gamma)e_2-(\beta-\delta\gamma)\omega_2}{3} - \frac{1}{3}(3+\alpha e_1 - \beta\omega_1)$. Letting $e_1 = e_2$ and $\omega_1 = \omega_2$, then $\Delta p_N = \frac{1}{3}\gamma\delta(-e+\omega)$.

Similarly, we obtain $\Delta p_S = \frac{1}{3}\gamma\delta(e-\omega)$, $\pi_N^V - \pi_N^{NV} = \frac{1}{18}\gamma\delta(e-\omega)(-6-2e\alpha+e\gamma\delta+2\beta\omega-\gamma\delta\omega) < 0$, and $\pi_{SO}^V - \pi_{SO}^{NV} = \frac{1}{18}\gamma\delta(1-\lambda)(e-\omega)(6-2e\alpha+e\gamma\delta+2\beta\omega-\gamma\delta\omega) > 0$.　□

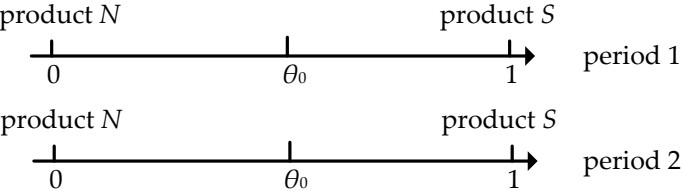

**Figure A2.** Variety-seeking consumer's utilities over two periods.

**Proof of Proposition 1.** Substituting the above equilibrium solutions into $\Delta e = e^V - e^{NV}$ and $\Delta\omega = \omega^V - \omega^{NV}$, we learn that $\Delta e$ is a long formula. Letting $\lambda = 0.2$, $k = 1$, $\alpha = 1$, and $\delta = 0.5$. Then we get $\Delta e = \frac{0.074\gamma(-42.11-3.722+\gamma^2)}{-36-4\gamma+\gamma^2} < 0$ and $\Delta\omega = \frac{\gamma(2.3+0.312\gamma-0.074\gamma^2)}{-36-4\gamma+\gamma^2} < 0$. Four prices of two products for two periods are shown in Figure 4a. According to $\Delta\pi_{SO} = \pi_{SO,2}^V - \pi_{SO,2}^{NV}$ and $\Delta\pi_N = \pi_{N,2}^V - \pi_{N,2}^{NV}$, we learn that $\Delta\pi_{SO}$ and $\Delta\pi_N$ are long formulas. Letting $\lambda = 0.2$, $k = 1$, $\alpha = 1$, $\delta = 0.5$, and $\gamma = 0.5$. Then we get $\Delta\pi_{SO} = 0.0190486 > 0$ and $\Delta\pi_N = 0.0068578 > 0$.　□

**Proof of Proposition 2.** We solve the partial derivatives of $e_2^V$ and $\omega_2^V$ with respect to $\gamma$, and obtain $\left|\frac{\partial e_2^V}{\partial\gamma}\right| - \left|\frac{\partial\omega_2^V}{\partial\gamma}\right| > 0$. Similarly, we obtain $\frac{\partial\pi_{N,2}^V}{\partial\gamma} - \frac{\partial\pi_{SO,2}^V}{\partial\gamma} < 0$.　□

**Proof of Proposition 3.** As shown in Appendix A Figure A2, $\theta_0 - \theta_1$ expresses the number of variety-seeking consumers who switch from buying traditional products to buying sharing products. $\theta_2 - \theta_0$ expresses the number of variety-seeking consumers who switch from buying sharing products to buying traditional products. Letting $\Delta q = (\theta_0 - \theta_1) - (\theta_2 - \theta_0)$ and substituting the above analytic solutions of $\theta_0$, $\theta_1$, and $\theta_2$, we obtain: $\Delta q = \frac{\gamma\delta((\alpha-\beta)(\alpha-\gamma\delta)(\beta-\gamma\delta)(-1+\lambda)-9k(\alpha+\beta)(-1+\lambda)-\gamma\delta\lambda)}{6k(18k-\alpha^2-\beta^2+2\alpha\gamma\delta+(\beta-\delta\gamma)(-\gamma\delta(-2+\lambda)+\beta\lambda))} > 0$.　□

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
