# Peer review of "Pricing and Service Level Decisions under a Sharing Product and Consumers’ Variety-Seeking Behavior"

_sustainability, doi:10.3390/su11246951_

Round 1

Reviewer 1 Report

The paper is interesting research investigated how having information about the consumers’ behaviors influence the demand in the sharing economy markets. The following sentences in the abstract are not clear:

“Furthermore, we found that the consumers’ variety-seeking behavior was not obvious, and the number of consumers shifting from the traditional product was monotonically increasing. While the variety-seeking behavior was obvious, the number of consumers shifting from the traditional product was monotonically decreasing.”

Did you mean: we found that when the consumers’ variety-seeking behavior was not obvious and the number of consumers shifting from the traditional product was monotonically increasing. Whereas when the variety-seeking behavior was obvious, the number of consumers shifting from the traditional product was monotonically decreasing.

The work needs editing and proofreading, the/a is missed at some points; also some of the plural/singular subjects/verbs should be checked. The introduction usually should provide answers to three main questions what is the problem? Why is the problem important? and how the article/research will deal with it? The introduction did not provide further information for the audience just a repetition of abstract and general information about the sharing economy. Paragraphs 171-178, I suggest the authors check their explanation of variables and equations. E.g. lines 171 and 174, full stops after S “i=N, S. u represents” is confusing you could revise your sentence to make less confusion for the audience. the concluding sentences that were previously used in the intro and abstract repeated in conclusion and with better phrasing. You could easily fix them in the intro and abstract. 

Author Response

See the enclosed.

Reviewer 2 Report

Literature review must be improved, especially for the application in the tourism sector.

Suggested readings:

Trijp, H. C. V., Hoyer, W. D., & Inman, J. J. (1996). Why Switch? Product Category–Level Explanations for True Variety-Seeking Behavior. Journal of marketing research33(3), 281-292.

Guttentag, D., Smith, S., Potwarka, L., & Havitz, M. (2018). Why tourists choose Airbnb: A motivation-based segmentation study. Journal of Travel Research57(3), 342-359.

Lin, P. M., Fan, D. X., Zhang, H. Q., & Lau, C. (2019). Spend less and experience more: Understanding tourists’ social contact in the Airbnb context. International Journal of Hospitality Management, 83, 65-73.

Tussyadiah, I. P., & Pesonen, J. (2016). Impacts of peer-to-peer accommodation use on travel patterns. Journal of Travel Research, 55(8), 1022-1040.

Legohérel, P., Hsu, C. H., & Daucé, B. (2015). Variety-seeking: Using the CHAID segmentation approach in analyzing the international traveler market. Tourism Management, 46, 359-366.

Kahn, B. E. (1998). Dynamic relationships with customers: High-variety strategies. Journal of the Academy of Marketing Science, 26(1), 45-53.

Zervas, G., Proserpio, D., & Byers, J. W. (2017). The rise of the sharing economy: Estimating the impact of Airbnb on the hotel industry. Journal of marketing research54(5), 687-705.

So, K. K. F., Oh, H., & Min, S. (2018). Motivations and constraints of Airbnb consumers: Findings from a mixed-methods approach. Tourism Management67, 224-236.

r. 50

“the same hotels (Hilton Worldwide) again”

Nowadays, Hilton is not any more a single brand-product group.

It owns 16 different lodging products.

See

In fact, brand-product portfolio management has become a primary issue for hotel groups.

And there has been a shift from standardization to differentiation.

Suggested readings:

Lim, Y. (2016). Brand management of hotel chains. Ivanova, M., Ivanov, S., & Magnini, V. P. (Eds.) The routledge handbook of hotel chain management, 221-229, Routledge.

Mauri, A. G. (2016). Pricing and revenue management in hotel chains. Ivanova, M., Ivanov, S., & Magnini, V. P. (Eds.)The Routledge Handbook of Hotel Chain Management, 262-273, Routledge.

Richard, B., & Cleveland, S. (2016). The future of hotel chains: Branded marketplaces driven by the sharing economy. Journal of Vacation Marketing22(3), 239-248.

r. 51

“Hotel Association of New York City 51 estimated that in the 12 months ending August 2015, Airbnb caused a direct loss of $451 billion for 52 New York City hotels”

insert updated figures

Pay attention to avoid same sentence of a similar publication.

Liu, Z., Feng, J., & Wang, J. (2019). Effects of the sharing economy on sequential innovation products. Complexity, 2019.

Insert a paragraph: Limitations of the study

Please, check again the mathematical model.

Author Response

See  the enclosed. 

Reviewer 3 Report

Dear authors,

First of all, I would like to congratulate you on focusing on this theme. It has a high level of novelty, especially by considering the importance of price in SE.

I have the following suggestion: after you established the two scenario, you had the analyses but, in my opinion, you did not exploited enough the results. A deeper discussion about the impact of the results on pricing decisions is recommended.

Author Response

See the enclosed.
